# Age-Dependent Distinct Distributions of Dendritic Cells in Autoimmune Dry Eye Murine Model

**DOI:** 10.3390/cells10081857

**Published:** 2021-07-22

**Authors:** Young-Ho Jung, Jin-Suk Ryu, Chang-Ho Yoon, Mee-Kum Kim

**Affiliations:** 1Department of Ophthalmology, College of Medicine, Seoul National University, 103 Daehak-ro, Jongno-gu, Seoul 03080, Korea; dudghmed86@naver.com (Y.-H.J.); ifree7@gmail.com (C.-H.Y.); 2Department of Ophthalmology, Seoul National University Hospital, 101 Daehak-ro, Jongno-gu, Seoul 03080, Korea; 3Laboratory of Ocular Regenerative Medicine and Immunology, Biomedical Research Institute, Seoul National University Hospital, 101 Daehak-ro, Jongno-gu, Seoul 03080, Korea; enter2357@naver.com; 4Transplantation Research Institute, Seoul National University Medical Research Center, 103 Daehak-ro, Jongno-gu, Seoul 03080, Korea

**Keywords:** aging, dendritic cells (DCs), antigen-presenting cell (APC), autoimmune dry eye, CD103 DC, MHC-II^hi^ B cell, CD11b DC

## Abstract

We investigated whether aging-dependent changes in dendritic cell (DC) distributions are distinct in autoimmune dry eye compared with an aging-related murine model. Corneal staining and tear secretion were evaluated in young and aged C57BL/6 (B6) and NOD.B10.H2b mice (NOD). In the corneolimbus, lacrimal gland (LG), and mesenteric lymph node (MLN), CD11b^−^ and CD11b^+^ DCs, CD103^+^ DCs and MHC-II^hi^ B cells were compared between young and aged B6 and NOD mice. With increased corneal staining, tear secretion decreased in both aged B6 and NOD mice (*p* < 0.001). In both aged B6 and NOD mice, the percentages of corneolimbal CD11b^+^ DCs were higher (*p* < 0.05) than those in young mice. While, the percentages of lymph nodal CD103^+^ DCs were higher in aged B6 and NOD mice (*p* < 0.05), the percentages of corneolimbal CD103^+^ DCs were only higher in aged NOD mice (*p* < 0.05). In aged NOD mice, the proportions of lacrimal glandial and lymph nodal MHC-II^hi^ B cells were also higher than those in young mice (*p* < 0.05). It indicates that corneolimbal or lacrimal glandial distribution of CD103^+^ DCs or MHC-II^hi^ B cells may be distinct in aged autoimmune dry eye models compared to those in aged immune competent murine models.

## 1. Introduction

Since Ralph Steinman and Zanvil Cohn discovered the dendritic cell (DC) [1], DCs have been studied as key mediators of innate and adaptive immunity [2,3]. DCs can also activate regulatory T cells (Treg) and tolerize T cells to self-antigens [4,5]. Thus, DCs are being studied for their potential therapeutic benefits against autoimmune diseases [6]. Specific DC subset functions change during aging or in a tissue-specific manner, and a breakdown in the balance between functions may lead to autoimmune diseases [7]. Therefore, changes in corneal or lacrimal glandial DCs during aging should be studied in dry eye models.

DCs are found both in lymphoid and in non-lymphoid organs such as the skin, muscle, lung, kidney, intestine, liver, and eye [1,2,8]. Several studies have identified heterogeneous populations of corneal DCs [8,9,10,11,12], which play immunogenic roles in dry eye syndrome, corneal allotransplant, and autoimmune diseases [12,13,14,15,16,17], or exhibit tolerogenic properties [18]. An imbalance in immunity contributes to the pathogenesis of dry eye diseases (DED) [19,20]. Given its worldwide prevalence, especially in elderly individuals [21,22], DED is a clinically relevant issue in today’s society. Recently, the kinetics and functions of DCs have been investigated in ocular surface inflammation with DED [23,24].

The characteristics of DCs in the ocular surface or lacrimal glands (LGs) have been investigated intensively in experimental dry eye models [14,23,25]. However, a study on the DC distributions in the cornea or LG associated with an autoimmune dry eye models is currently under-investigated. Although existing studies provide some evidence for the role of intestinal dysbiosis in autoimmune dry eyes [26,27,28], there are no pertinent studies on DED directly linking changes in DCs with the ocular–gut axis. Therefore, we investigated whether the distribution of DC subsets is distinct in aged NOD.B10.H2^b^ (NOD) mice that mimic Sjögren’s syndrome-like changes when compared to the distribution in aged C57BL/6 (B6) mice [26]. The study also aimed to uncover whether the difference in distribution is also accompanied by changes in DC subsets of mesenteric lymph nodes (MLNs) as a method to link the variation in DCs to the ocular–gut axis.

## 2. Materials and Methods

### 2.1. Animals

Eight C57BL/6 (B6) male mice and eight NOD.B10.H2^b^ male mice (The Jackson Laboratory, Bar Harbor, ME) were used in this study. Eight-week- (*n* = 4) and 20-month-old (*n* = 4) B6 mice were included as the young and aged B6 group, while five-week- (*n* = 4) and 24-week-old (*n* = 4) NOD.B10.H2^b^ mice were included as the young and aged NOD group. The mice were bred in a specific pathogen-free facility at the Biomedical Research Institute of Seoul National University Hospital (Seoul, Korea), maintained at 22–24 °C with 55 ± 5% relative humidity, and provided free access to food and water.

### 2.2. Clinical Evaluation of the Dry Eye

Corneal staining and tear secretion tests were performed under anesthesia (using a mixture of zoletil and xylazine at a ratio of 1:3). Corneal staining was blindly assigned by a single experienced ophthalmologist (Y. J.) using the National Eye Institute (NEI) scoring scheme. Fluorescein dye (0.25%) and Lissamine Green B (3%) (Sigma-Aldrich, St. Louis, MO, USA) were used for corneal staining in B6 and NOD.B10.H2^b^ mice, respectively. The different dyes were selected for visualizing dry eyes in the two types of mice both having different levels of iris pigments, as reported in previous studies [29,30,31,32].

After placing one drop of dye on the conjunctival sac for 30 seconds, the ocular surface was gently washed with 1 mL of normal saline. Corneal staining was observed using a microscope (Olympus SZ61; Olympus Corporation, Tokyo, Japan). The B6 and NOD.B10.H2b mice were observed using cobalt blue and white light (LED) illumination, respectively [26,33]. For the tear secretion test, phenol red-impregnated cotton threads (FCI Ophthalmics, Pembroke, MA, USA) were placed into the lateral canthus of mice for 60 s. The average values of corneal staining scores and tear secretion results from both eyes of each mouse were used for the statistical analyses based on a protocol described in a previous study [34]. Tear volume was adjusted by body weight for the analysis based on protocols used in previous studies because LG increased with aging and a positive correlation between body weight and LG size has been reported [32,35,36,37].

### 2.3. Flow Cytometry for Analyzing Immune Cells

After the mice were euthanized, the cornea and limbal tissue (Co), extraorbital LG, and mesenteric lymph node (MLN) were extracted and collected. Bilateral corneolimbal tissues were pooled in each mouse because the number of immune cells in unilateral corneolimbal tissue is not sufficient to conduct flow cytometric analysis [32]. Whereas, a lacrimal gland was used in each mouse without pooling. To obtain a single-cell suspension, the tissues were treated with 1 mg/mL collagenase type I (Worthington, Lakewood, NJ, USA) for 30 min at 37 °C and homogenized through a 70-μm filter (BD BioSciences, San Diego, CA, USA).

A single cell suspension was prepared by mincing the tissue between the frosted ends of two glass slides in RPMI-1640 medium (WelGENE, Daegu, Korea), 10% fetal bovine serum (FBS), and 1% penicillin-streptomycin. To sort dendritic cells (DCs), antigen-presenting MHC-II^hi^ B cells, macrophages, and granulocytes, including neutrophils and eosinophils, the cell populations were identified using the sequential gating strategy (Figure 1). To exclude non-immune cells in Co and LG, the cells were gated with CD45 and the percentage of APC subsets was calculated in the CD45^+^ parent population. The following antibodies were used for a subset: CD11b-PE-Cy7 (Cat no. 25-0112, 0.125 μg/test; eBioscience, San Diego, CA, USA), CD11c-PerCP-Cy5.5 (Cat no. 45-0114, 0.25 μg/test; eBioscience), MHC-II-FITC (Cat no. 11-5321, 0.125 μg/test; eBioscience), CD24-PE (Cat no. 12-0242, 0.03 μg/test; eBioscience), and CD45-APC (Cat no. 17-0451, 0.125 μg/test; eBioscience). To identify CD103+ dendritic cells (cDC), CD103-PE (Cat no. 12-1031, 1 μg/test; eBio-science) was applied to cells stained with CD11b-PE-Cy7, CD11c-PerCP-Cy5.5, MHC-II-FITC, and CD45-APC in the other subset. The stained cells were assayed using a FACSCanto flow cytometer (BD BioSciences, San Jose, CA, USA). Data were analyzed using FlowJo software (version 10.7.1) (Tree Star, Ashland, OR, USA).

After forward-scatter and side-scatter gating, CD45^+^ cells were gated (Figure 1). The cells were then further divided by CD11b and CD11c expression. Thereafter, each proportion was sorted by the expression of CD24 and MHC-II or by the expression of CD103 and MHC-II (Figure 1). Antigens presenting MHC-II^hi^ B cells were identified with CD45^+^CD11b^−^CD11c^−^CD24^+^MHC-II^hi^ (Figure 1A) [38]. Antigen-presenting MHC-II^hi^ macrophages (MHC-II^hi^ macrophage) or non-antigen-presenting macrophages were identified as CD45^+^CD11b^+^CD11c^−^CD24^−^MHC-II^hi^ or CD45^+^CD11b^+^CD11c^−^CD24^−^MHC-II^lo^ (Figure 1B,C) [39]. Within CD11b^+^ cells, neutrophils/eosinophils were isolated with CD45^+^CD11b^+^CD11c^−^CD24^+^MHC-II^lo^ (Figure 1D) [39]. Based on the expression of CD45, CD11c, CD24, and MHC-II as dendritic cells (DCs), DCs were further divided into CD11b^−^ DCs and CD11b^+^ DCs (Figure 1E,F) [39]. CD103^+^CD11b^−^ DCs and CD103^+^CD11b^+^ DCs were gated based on CD45^+^CD11b^−^CD11c^+^CD103^+^MHC-II^hi^ and CD45^+^CD11b^+^CD11c^+^CD103^+^MHC-II^hi^ (Figure 1G,H) [40].

### 2.4. Statistical Analyses

For comparing the two groups, independent t-test was used. Statistical analyses were performed using GraphPad Prism software (version 9.0; GraphPad Software, La Jolla, CA, USA). Differences were considered statistically significant at *p* < 0.05. Data was expressed as mean ± SD for all experimental measurements.

## 3. Results

### 3.1. Dry Eye Is Manifest in Aged B6 and NOD.B10.H2^b^ Mice

In immunocompetent B6 mice, the average corneal stain scores of aged mice were significantly higher (13.63 ± 1.60) than those of young mice (4.75 ± 4.84) (*p* = 0.013; Figure 2A). Tear secretion was not significantly different between the groups (10.85 ± 2.78 mm in young and 7.45 ± 4.01 mm in aged mice) (*p* = 0.212). However, when body weight (BW) was adjusted, the corrected value was significantly lower in aged mice (0.17 ± 0.09/BW) than in young mice (0.48 ± 0.13/BW) (*p* = 0.008; Figure 2B).

In autoimmune-susceptible NOD.B10.H2^b^ mice, the corneal stain scores of aged mice were significantly higher (10.50 ± 2.04) than those of young mice (2.88 ± 2.25) (*p* = 0.002; Figure 2C). A marked reduction in tear secretion was observed in aged mice. The tear secretion in young and aged mice was 4.86 ± 0.37 mm and 3.44 ± 0.30 mm, respectively (*p* = −0.001). BW adjusted tear secretion was also significantly lower in the aged group (0.11 ± 0.01/BW) than in the young group (0.19 ± 0.02/BW) (*p* < 0.001; Figure 2D).

### 3.2. Changes of Distribution CD11b^+^DC Are Distinct in Aged Mice

The percentages of the CD45^+^ cell in corneolimbus, LG, and MLN showed no differences between the young and aged mice. As shown in Figure 3, the percentage of corneolimbal CD11b^+^ DCs (2.53 ± 0.29%) was significantly higher in aged B6 mice than in young B6 mice (1.43 ± 0.38%) (*p* = 0.004). The proportions of MLN CD11b^+^ DCs was also higher in aged B6 mice (4.27 ± 1.07%) than in young B6 mice (2.82 ± 0.09%) (*p* = 0.035; Figure 3A), which corresponded with the changes of the corneolimbal CD11b^+^ DCs. Unlike the corneolimbal and nodal DC distributions, lacrimal glandial CD11b^+^ DCs (18.26 ± 4.14%) in aged mice was not increased compared to those in young mice (25.28 ± 3.65%; (Figure 3A). Similar to aged B6 mice, the percentage of corneolimbal CD11b^+^ DCs in aged NOD.B10.H2^b^ mice was also higher (6.96 ± 1.85%) than in young mice (0.54 ± 0.48%) (*p* < 0.001; Figure 3B).

### 3.3. Changes of Distribution CD103^+^CD11b^−^DC Subsets in Aged Mice

There was no significant proportions difference of the CD45^+^ cells in corneolimbal, LG and MLN between the young and aged mice. No significant changes were revealed in B6 mice (Figure 4A). However, notably, the percentages of MLN CD103^+^CD11b^−^ DCs (1.29 ± 0.17%) was significantly higher in aged NOD.B10.H2^b^ mice than in young mice (0.74 ± 0.18%, *p* = 0.004; Figure 4B). Correspondingly, the percentage of corneolimbal CD103^+^CD11b^−^ DCs was markedly higher in aged mice (8.13 ± 1.63%) than in young mice (0.25 ± 0.29%) (*p* < 0.001; Figure 4B).

### 3.4. Changes of Distribution CD103^+^CD11b^+^DC Subsets in Aged Mice

Although the percentages of MLN CD103^+^CD11b^+^ DCs were increased in the aged B6 mice (0.89 ± 0.03% vs. 1.74 ± 0.29%, *p* < 0.05; Figure 5A), no differences were observed in corneolimbal or lacrimal glandial CD103^+^ DC subsets (Figure 5A). Correspondingly, the percentages of MLN CD103^+^CD11b^+^ DCs (0.32 ± 0.07%) was significantly higher in aged NOD.B10.H2^b^ mice than in young mice (0.11 ± 0.05% *p* = 0.002; Figure 5B).

### 3.5. Distributions of MHC-II^hi^ B Cells in Aged Mice

In NOD.B10.H2^b^ mice, the percentages of lacrimal glandial MHC-II^hi^ B cells were significantly higher in aged NOD.B10.H2^b^ mice (11.70 ± 0.36%) compared to young NOD.B10.H2^b^ mice (7.84 ± 2.52%) (*p* = 0.023; Figure 6B), which was accompanied by the increased proportion of MLN MHC-II^hi^ B cells in the aged NOD.B10.H2^b^ mice (1.96 ± 0.42% vs. 1.30 ± 0.26% in young mice, *p* = 0.038; Figure 6B). Unlike NOD.B10.H2^b^ mice, there was no differences in proportion of MHC-II^hi^ B cells in aged B6 mice (Figure 6A).

### 3.6. Distributions of Other Antigen Presenting Cell (APC) Subsets in Aged Mice

The proportional changes of corneolimbal MHC-II^hi^ macrophage or neutrophils/eosinophils were not observed in two dry eye mouse models (Figure 7 and Figure 8). The percentage of MHC-II^hi^ macrophage revealed considerable decrease in old B6 mice (1.12 ± 0.13%) than young B6 mice (2.03 ± 0.21%) at LG (*p* < 0.05, Figure 7A). The percentages of lacrimal glandial MHC-II^hi^ macrophage were also decreased in the aged NOD.B10.H2^b^ mice (MHC-II^hi^ macrophage; 3.41 ± 0.64 vs. 1.50 ± 0.03, *p* < 0.05, Figure 7B), although the proportion of nodal MHC-II^hi^ macrophage was increased in aged NOD.B10.H2^b^ mice (MHC-II^hi^ macrophage; 0.00 ± 0.00 vs. 0.002 ± 0.00, *p* < 0.05, Figure 7B).

The percentage of MLN neutrophils/eosinophils was significantly higher in aged B6 mice (1.04 ± 0.15) than young B6 mice (0.62 ± 0.03%, *p* < 0.05, Figure 8A).

In aged NOD.B10.H2^b^ mice, the percentages of lacrimal glandial neutrophils/eosinophils were decreased (neutrophil/eosinophil; 0.36 ± 0.05 vs. 0.19 ± 0.03, *p* < 0.05, Figure 8B), although the proportion of nodal neutrophils/eosinophils were increased in aged mice (neutrophil/eosinophil; 0.008 ± 0.00 vs. 0.038 ± 0.01, *p* < 0.05, Figure 7 and Figure 8B).

Even though they were statistically significant, fractions were too low and it is considered unlikely as biologically relevant.

## 4. Discussions

To the best of our knowledge, this is the first descriptive study on how distinct dendritic cell subsets are in immune susceptible aged mice when compared with those in immune competent aged mice. This study indicates that CD103^+^CD11b^−^ DCs and MHC-II^hi^ B cells may be the prevailing APCs in the ocular surface–LG axis of aged immune susceptible mice, whereas changes in CD103^+^ DCs are not recognizable in aged immune competent mice. Secondly, increased corneolimbal CD11b^+^ DC subsets are found in both aged B6 and NOD mice. It suggests that CD11b^+^ DC subsets may be one of the culprit cells for age-dependent ocular surface inflammation regardless of immune susceptible or competent status.

We focused on the changes in three subsets of DCs, namely, CD11b^+^ DCs, CD11b^−^ DCs, and CD103^+^ DCs [8,41]. CD103^+^ DCs act in self or foreign antigen recognition or in the induction of gut-related molecules on effector T cells [41]. IFN-α-producing plasmacytoid DCs are pathogenic cells that enhance pro-inflammatory cytokine production in Sjögren’s syndrome [42,43]. CD11c^+^CD86^+^ and CD11b^+^CD11c^−^ cells are involved in experimental dry eye models [14,23]. B cells detect and initiate their immune responses to antigens and present antigens to T cells [44,45,46]. Given that B cells are crucial for the pathogenesis of Sjögren’s syndrome [47], some DC subsets or antigen-presenting B cells may be responsible for aggravating the inflammatory responses in either aged immune competent B6 or autoimmune Sjögren’s syndrome-like mice.

Heterogeneous groups of APCs exist in the cornea and [17,48] have been investigated for their role in different inflammatory corneal diseases [49,50,51,52]. In this study, CD11b^+^ DCs were found to increase on the ocular surface in both aged B6 and NOD mice. Among the APCs, cDCs that express high levels of CD11c and MHC-II and low levels of F4/80 exhibited a superior capacity to present antigens to T cells [53]. cDC can be further sub-classified into type 1 cDC (cDC1) and type 2 cDC (cDC2) [53]. The former induces cytotoxic CD8^+^ T cell and T helper1 (Th1) responses, while the latter initiates Th2, Th17, and T regulatory (Treg) responses [53]. cDC1 can be identified as CD4^−^CD8α^+^CD11b^−^CD11c^+^ and cDC2 can be identified as CD4^+^CD8α^−^CD11b^+^CD11c^+^. In the eye, CD11b^+^ DC is known to have a crucial role in dry eye or post-keratoplasty rejection [10,14,54,55,56].

Corresponding with previous studies [10,14,54,55,56], this study supports that CD11b^+^DC may be an important subset in age-related ocular surface inflammation, by showing concomitant increase of corneolimbal CD11b^+^ DCs in aged mice, regardless of immune competency.

In gut nonlymphoid tissues, cDC1s are classified as CD103^+^CD11b^−^CD11c^+^MHC-II^hi^, whereas cDC2s comprise both CD103^+^CD11b^+^CD11c^+^MHC-II^hi^ and CD103^−^CD11b^+^CD11c^+^MHC-II^hi^ [53]. Surprisingly, CD103^+^CD11b^−^ DCs are remarkably increased in both the corneolimbus and MLN in aged NOD mice, suggesting an increase in the number of presumed CD103^+^ cDC1s. CD45^+^CD11b^−^CD11c^+^CD24^+^MHC-II^hi^ cells or CD45^+^CD11b^+^CD11c^+^CD24^+^MHC-II^hi^ cells are considered as cDC1 (CD11b^+^ DC) or cDC2 (CD11b^−^ DC). However, further detailed identification is needed using additional markers, such as CD8α and CD4. Nevertheless, this outcome provides an insight into how the distribution of CD11b^+^ DCs or CD103^+^ DCs changes in the corneolimbus and LG depending on age in either immune competent or immune susceptible murine models. In the present study, the mechanisms underlying the changes in the matured status of each type of APC were not evaluated. Given that the different distributions and maturity of APCs may have an effect on triggering immune responses, the different status of matured DCs should be investigated further using additional markers, namely, CD86 or CD80.

There is a distinct difference between the aging-related and autoimmune-related pathogenesis of dry eyes. Recent evidence indicates that B cells, Th1, Th2, and Th17 cells play major roles in the pathogenesis of Sjögren’s syndrome [14,47,57,58]. NOD.B10-H2^b^ mice exhibit many features of Sjögren’s syndrome, including exocrine gland dysfunction concomitant with leukocyte infiltration of the salivary gland and LGs [57]. Recent studies have shown that both Sjögren’s syndrome patients and NOD.B10-H2^b^ mice exhibit increased B lymphocyte survival, B cell hyper-reactivity, and hyper-gammaglobulinemia with a high production of autoantibodies [57]. In our study, antigen-presenting MHC-II^hi^ B cells tended to increase in both LG and MLN with aging in NOD.B10-H2^b^ mice, suggesting that B cells may be an important factor, not only as an effector cell to produce antibodies, but also as an APC in autoimmune dry eye. On the other hand, no changes in MHC-II^hi^ B cells were observed in aged B6 mice, indicating that antigen-presenting B cells may not to be involved in the pathogenesis of immune competent dry eye. This change in the lacrimal glandial DCs is distinct from the change in corneolimbal DCs, suggesting that heterogeneous aging-dependent distributions of DC or other APCs vary depending on the tissues. Given that gut dysbiosis has a significant impact on the maturation and differentiation of B cells [59], an increase in mesenteric nodal MHC-II^hi^ B cells may be gut-related. Considering that recent studies have shown that intestinal dysbiosis is related to autoimmune dry eyes [26,27,28], the increase in lacrimal glandial MHC-II^hi^ B cells observed in the present study may correspond to changes in the gut. In primary Sjögren’s syndrome, CD27^+^ memory B cells, marginal zone B cells, plasmablasts, and plasma cells are the key subsets of B cells [47]. To acknowledge the functional changes of effector B cells, further investigation should be performed using CD19, CD20, and CD38, or measured using antibodies [60]. Th1 and Th17 cells and related cytokines are known to be involved in age-related dry eyes [22,32,61]. The role of B cells in age-related dry eyes has been disputed [22,61]. This study suggests that T cells are related to dry eye pathogenesis in both aged and autoimmune mice by showing an increase in CD11b^+^ DC subsets in both mice. However, the increase in MHC-II^hi^ B cells is only shown in immunosusceptible NOD mice, supporting results observed in previous studies [47,57].

This study has several limitations. First, only male mice were used because female NOD.B10.H2^b^ mice tend to develop only sialadenitis instead of dacryoadenitis [62]. Only B6 male mice were included to match the sex with NOD.B10.H2^b^ mice. However, the female sex is one of the highest risk factors for dry eye syndrome, due to the effects of hormones in this disorder [63]. Therefore, it is not possible to apply female-associated dry eye. Second, the sample size was low since only a few animals were used. NOD.B10.H2^b^ mice have low reproductive rates and only male NOD.B10.H2^b^ mice can be used since dacryoadenitis occurs only in males. Additionally, aged B6 mice have low rates of survival, hence the low sample size can be justified with these limitations. Several previous studies also included a small sample size owing to the rarity of experimental animals [26,64,65]. We analyzed the data from as many mice as possible for analysis. However, at the time of the experiment, only four male 20-month-old B6 mice and four male 24-week-old NOD.B10.H2^b^ mice were available. Third, a flow cytometry machine capable of sorting up to five channels was used. Therefore, the use of additional specific markers to differentiate between neutrophil and eosinophil, cDC1 and cDC2, or mature DC and immature DCs was not possible [66]. However, a comprehensive study of the DC subtypes in the corneolimbus, LG, and MLNs of an aging-dependent murine dry eye model is lacking. Therefore, this work sheds light on the distinct age-dependent distributions of APCs in different tissues, such as in the corneolimbus or LG, in aging-related dry eye disease.

## 5. Conclusions

This study demonstrates that there is a distinct heterogeneous distribution of corneolimbal or lacrimal glandial DCs between immune competent and immune susceptible murine models. In aged Sjögren’s syndrome-like mice, an increase in corneolimbal CD103^+^ DCs and lacrimal glandial MHC-II^hi^ B cells is distinctively observed which is accompanied by an increase in nodal CD103^+^ DCs and MHC-II^hi^ B cells, whereas increase of corneolimbal CD11b^+^ DCs is concomitantly found in the aged murine model, irrespective of immune competency.

## Figures and Tables

**Figure 1 cells-10-01857-f001:**
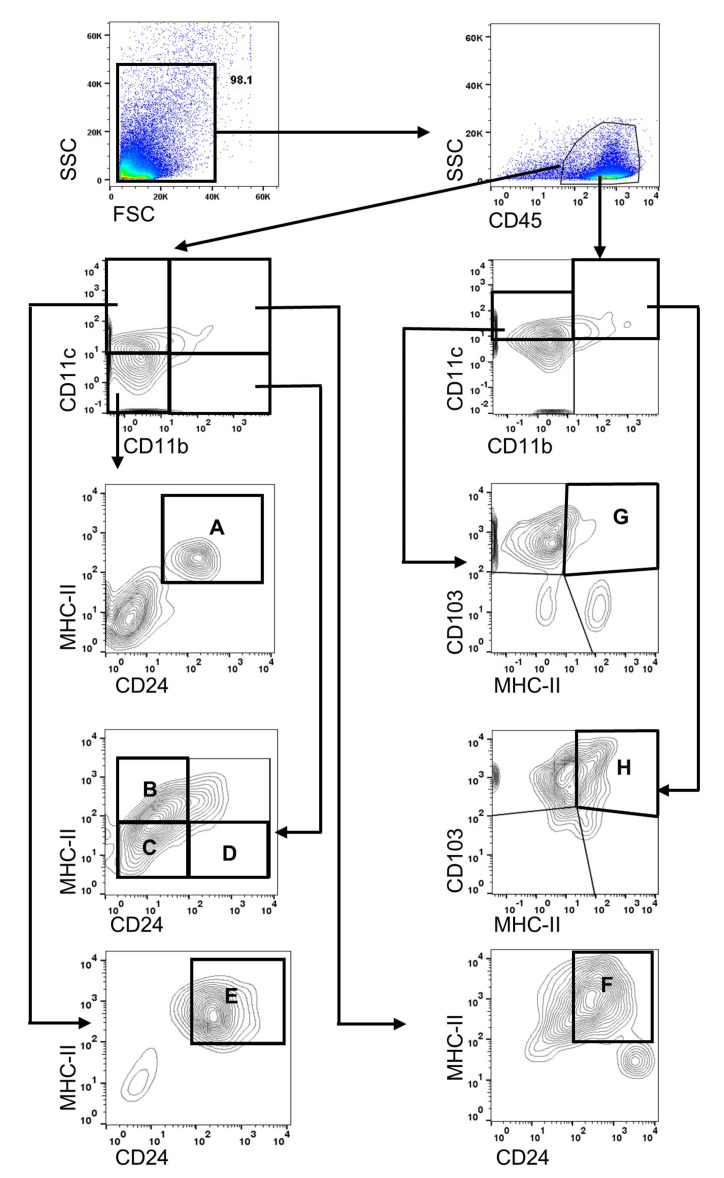
Gating strategy used to identify myeloid-cell and lymphoid-cell subsets. Following the exclusion of debris and cellular aggregates, forward-scatter and side-scatter gating were performed. The pan-hematopoietic marker CD45 was selected to distinguish immune cells from other corneolimbal or lacrimal gland cells. A sequential gating strategy was used to identify populations expressing specific markers: (**A**) MHC-II^hi^ B cells, (**B**) MHC-II^hi^ macrophages, (**C**) MHC-II^lo^ macrophages, (**D**) neutrophils and eosinophils, (**E**) CD11b^-^DC, (**F**) CD11b^+^ DC, (**G**) CD103^+^CD11b^-^DC, and (**H**) CD103^+^CD11b^+^ DC. DC, dendritic cells.

**Figure 2 cells-10-01857-f002:**
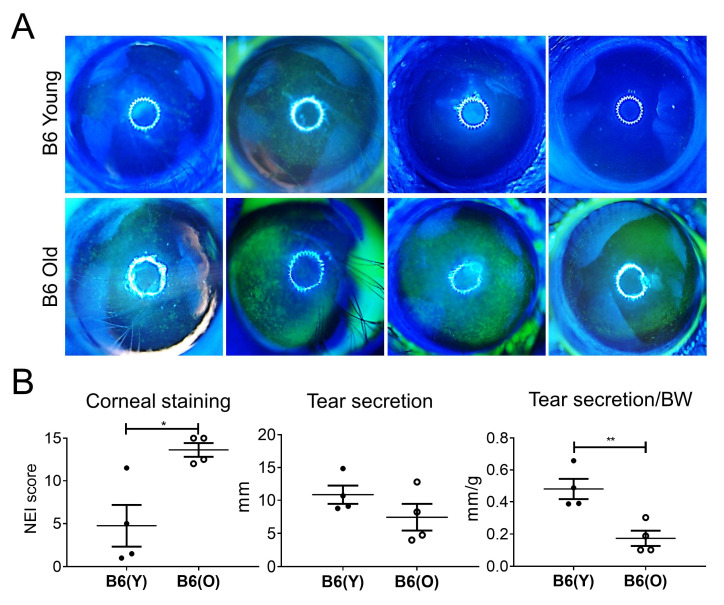
(**A**) Representative images of corneal fluorescein staining of young C57BL/6 (B6) mice (8-week-old; upper row, *n* = 4) and aged B6 mice (20-month-old; lower low, *n* = 4). (**B**) The National Eye Institute (NEI) corneal staining score was significantly higher in aged B6 mice than in young B6 mice (*p* = 0.013). Body weight (BW) adjusted tear secretion was significantly lower in aged B6 mice than young B6 mice (*p* = 0.008). (**C**) Representative images of corneal lissamine green staining of young NOD.B10.H2^b^ mice (5-week-old; upper row) and aged NOD.B10.H2b mice (24-week-old; lower row). (**D**) The NEI corneal staining score was significantly higher in aged NOD.B10.H2^b^ mice than in young NOD.B10.H2^b^ mice (*p* = 0.002). Tear secretion and BW-adjusted tear secretion were significantly lower in aged NOD.B10.H2^b^ mice than in young NOD.B10.H2^b^ mice (*p* < 0.001). Clinical scores were analyzed using unpaired, two-tailed, independent t-test. Data are expressed as the mean ± standard error. Each dot represents average values of either corneal staining scores or tear secretion results from both eyes in each mouse. B6(Y), young B6 mice; B6(O), aged B6 mice; NOD(Y), young NOD.B10.H2^b^ mice; NOD(O), aged NOD.B10.H2^b^ mice. * *p* < 0.05, ** *p* < 0.01, *** *p* < 0.001.

**Figure 3 cells-10-01857-f003:**
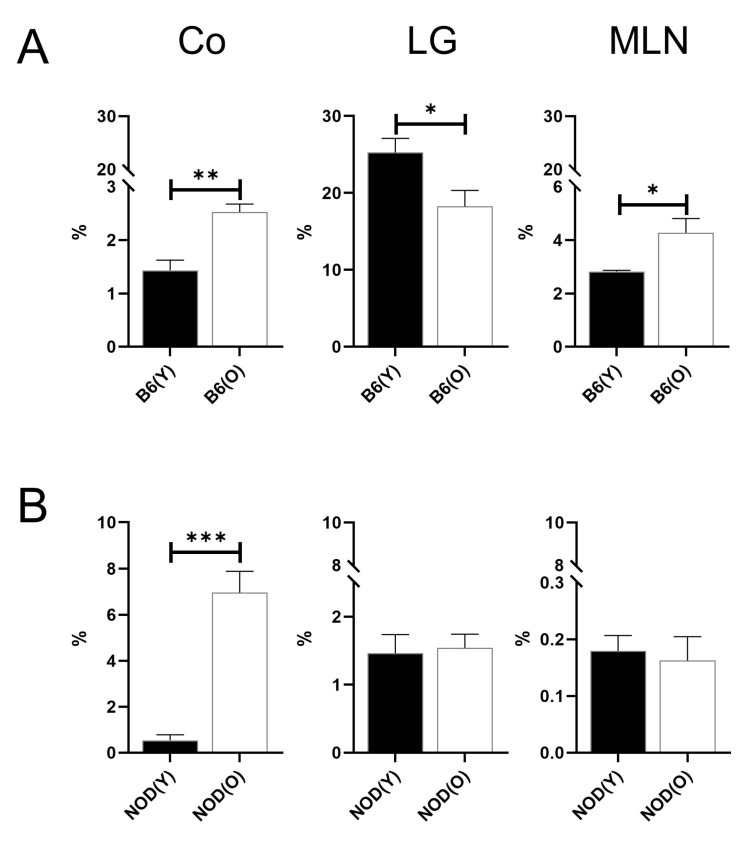
Distribution of CD11b^+^ DCs in B6 (**A**) (*n* = 4) and NOD.B10.H2^b^ (**B**) mice (*n* = 4). The total number of CD45^+^ cells was used as the denominator to compensate for the actual fraction of each subset in corneolimbus and LG. In MLN, the percentage of subsets was calculated as the percentage within total cells. The percentage of CD11b^+^ DCs was higher in corneolimbus and MLN of aged B6 mice (*p* = 0.004, *p* = 0.035, respectively). However, CD11b^+^ DCs of LG were significantly lower than in young B6 mice (*p* = 0.044). The percentage of corneolimbal CD11b^+^ DCs in aged NOD.B10.H2^b^ mice was significantly higher than in young NOD.B10.H2^b^ mice (*p* < 0.001). All experiment results were tested for statistical differences using unpaired, two-tailed independent t-tests. Data are expressed as the mean ± standard error of the mean. B6, C57BL/6; NOD, NOD.B10.H2^b^; DC, dendritic cell; CO, corneolimbus; LG, lacrimal gland; MLN, mesenteric lymph node; B6(Y), young B6 mice; B6(O), aged B6 mice; NOD(Y), young NOD.B10.H2^b^ mice; NOD(O), aged NOD.B10.H2^b^ mice. * *p* < 0.05, ** *p* < 0.01, *** *p* < 0.001.

**Figure 4 cells-10-01857-f004:**
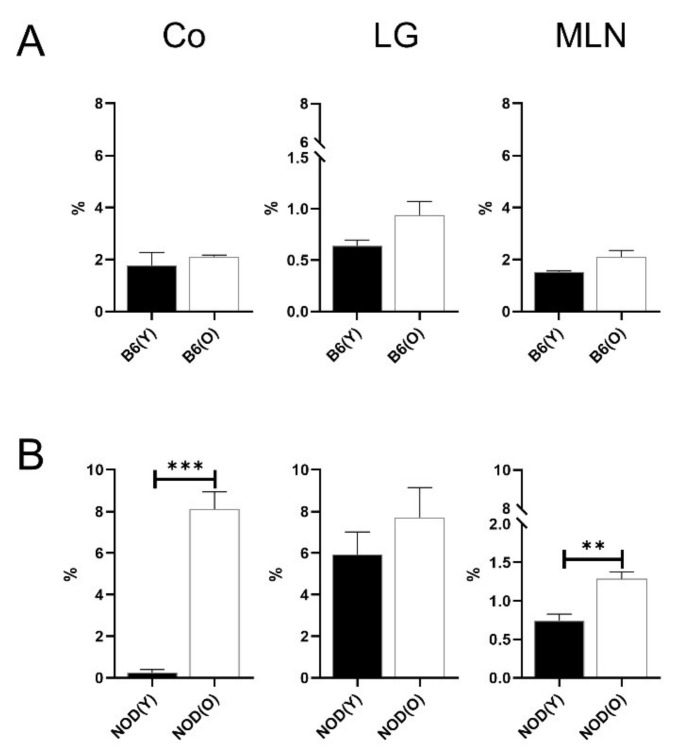
Distribution of CD103^+^CD11b^−^ DCs in B6 (**A**) (*n* = 4) and NOD.B10.H2^b^ (**B**) mice (*n* = 4). The total number of CD45^+^ cells was used as the denominator to compensate for the actual fraction of each subset in corneolimbus and LG. In MLN, the percentage of subsets was calculated as the percentage within total cells. The percentage of MLN CD103^+^CD11b^−^ DCs was significantly higher in aged NOD.B10.H2^b^ mice than in young NOD.B10.H2^b^ mice (*p* = 0.004). Correspondingly, the percentage of corneolimbal CD103^+^CD11b^−^ DCs was markedly higher in aged NOD.B10.H2^b^ mice than in young NOD.B10.H2^b^ mice (*p* < 0.001). All experiment results were tested for statistical differences using unpaired, two-tailed independent *t*-tests. Data are expressed as the mean ± standard error of the mean. B6, C57BL/6; NOD, NOD.B10.H2^b^; DC, dendritic cell; CO, corneolimbus; LG, lacrimal gland; MLN, mesenteric lymph node; B6(Y), young B6 mice; B6(O), aged B6 mice; NOD(Y), young NOD.B10.H2^b^ mice; NOD(O), aged NOD.B10.H2^b^ mice. ** *p* < 0.01, *** *p* < 0.001.

**Figure 5 cells-10-01857-f005:**
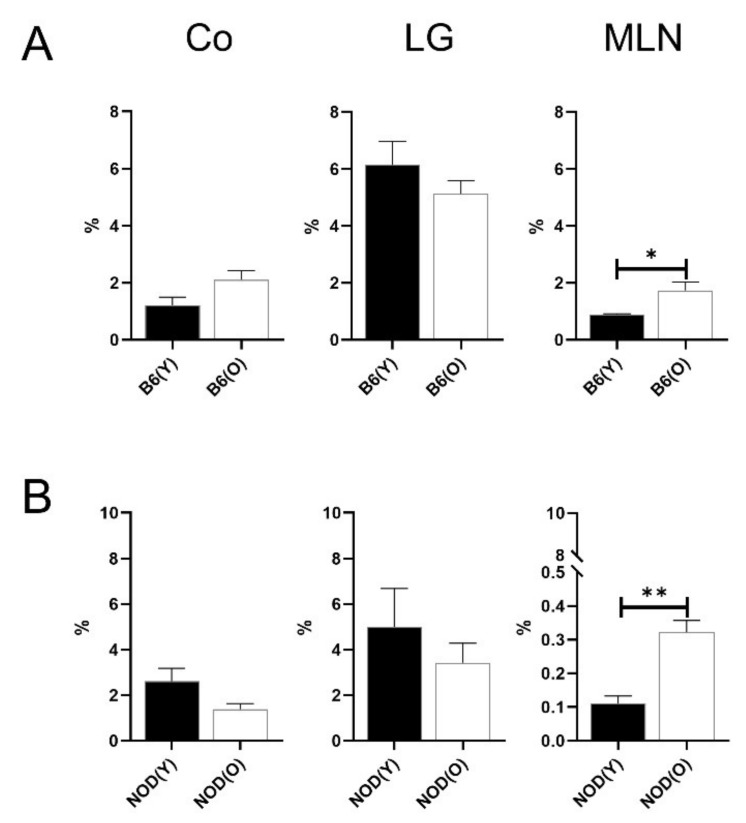
Distribution of CD103^+^CD11b^+^ DCs in B6 (**A**) (*n* = 4) and NOD.B10.H2^b^ (**B**) mice (*n* = 4). The total number of CD45^+^ cells was used as the denominator to compensate for the actual fraction of each subset in corneolimbus and LG. In MLN, the percentage of subsets was calculated as the percentage within total cells. The percentage of MLN CD103^+^CD11b^+^ DCs were significantly increased in both mouse models (*p* = 0.025 in B6, *p* = 0.002 in NOD.B10.H2^b^, respectively). All experiment results were tested for statistical differences using unpaired, two-tailed independent t-tests. Data are expressed as the mean ± standard error of the mean. B6, C57BL/6; NOD, NOD.B10.H2^b^; DC, dendritic cell; CO, corneolimbus; LG, lacrimal gland; MLN, mesenteric lymph node; B6(Y), young B6 mice; B6(O), aged B6 mice; NOD(Y), young NOD.B10.H2^b^ mice; NOD(O), aged NOD.B10.H2^b^ mice. * *p* < 0.05, ** *p* < 0.01.

**Figure 6 cells-10-01857-f006:**
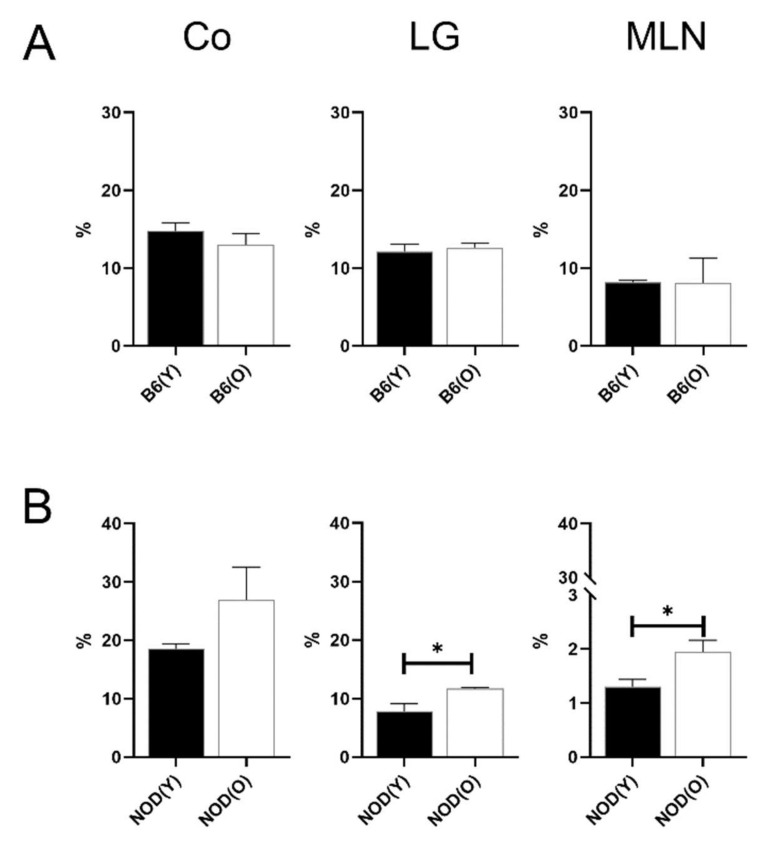
Distribution of MHC-II^hi^ B cells in B6 (**A**) (*n* = 4) and NOD.B10.H2^b^ (**B**) mice (*n* = 4). The total number of CD45^+^ cells was used as the denominator to compensate for the actual fraction of each subset in corneolimbus and LG. In MLN, the percentage of subsets was calculated as the percentage within total cells. No significant proportional changes were observed in the MHC-II^hi^ B cell in the corneolimbus, LG, and MLN of B6 mice. The percentage of lacrimal glandial MHC-II^hi^ B cells was significantly higher in aged NOD.B10.H2^b^ mice compared to young NOD.B10.H2^b^ mice (*p* = 0.023). The percentage of MLN MHC-II^hi^ B cells was significantly higher in the aged NOD.B10.H2^b^ mice than in the young NOD.B10.H2^b^ mice (*p* = 0.038). All experiment results were tested for statistical differences using unpaired, two-tailed independent t-tests. Data are expressed as the mean ± standard error of the mean. B6, C57BL/6; NOD, NOD.B10.H2^b^; DC, dendritic cell; CO, corneolimbus; LG, lacrimal gland; MLN, mesenteric lymph node; B6(Y), young B6 mice; B6(O), aged B6 mice; NOD(Y), young NOD.B10.H2^b^ mice; NOD(O), aged NOD.B10.H2^b^ mice. * *p* < 0.05.

**Figure 7 cells-10-01857-f007:**
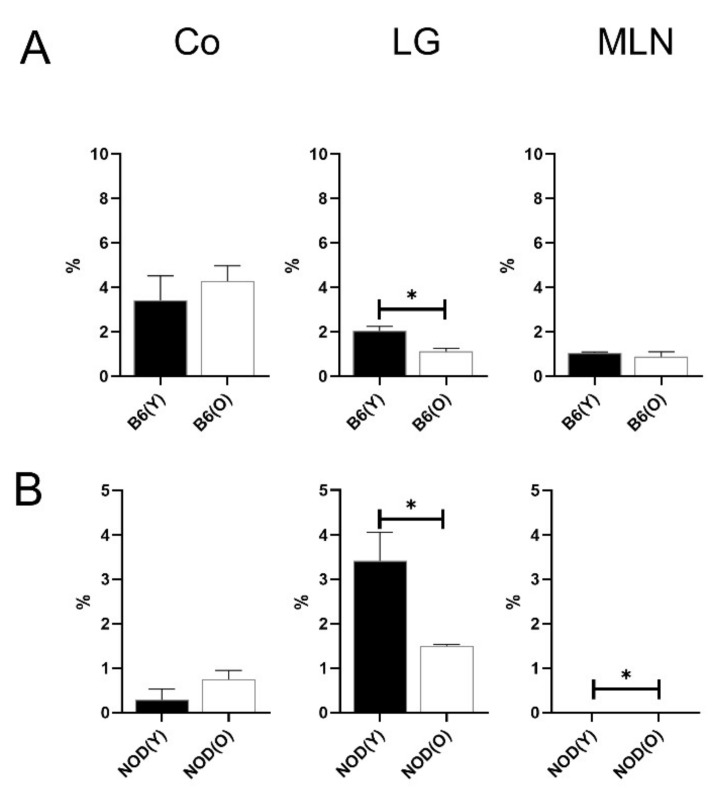
Distribution of macrophages in B6 (**A**) (*n* = 4) and NOD.B10.H2^b^ (**B**) mice (*n* = 4). The total number of CD45^+^ cells was used as the denominator to compensate for the actual fraction of each subset in corneolimbus and LG. In MLN, the percentage of subsets was calculated as the percentage within total cells. The percentage of lacrimal glandial MHC-II^hi^ macrophage (CD11b^+^CD11c^−^CD24^−^MHC-II^hi^) was significantly lower in aged than young mice in both mice (*p* = 0.010 in B6, *p* = 0.024 in NOD.B10.H2^b^, respectively). In contrast, MLN showed a considerably increased proportion of MHC-II^hi^ macrophage in aged NOD.B10.H2^b^ mice (*p* = 0.039). All experiment results were tested for statistical differences using unpaired, two-tailed independent t-tests. Data are expressed as the mean ± standard error of the mean. B6, C57BL/6; NOD, NOD.B10.H2^b^; DC, dendritic cell; CO, corneolimbus; LG, lacrimal gland; MLN, mesenteric lymph node; B6(Y), young B6 mice; B6(O), aged B6 mice; NOD(Y), young NOD.B10.H2^b^ mice; NOD(O), aged NOD.B10.H2^b^ mice. * *p* < 0.05.

**Figure 8 cells-10-01857-f008:**
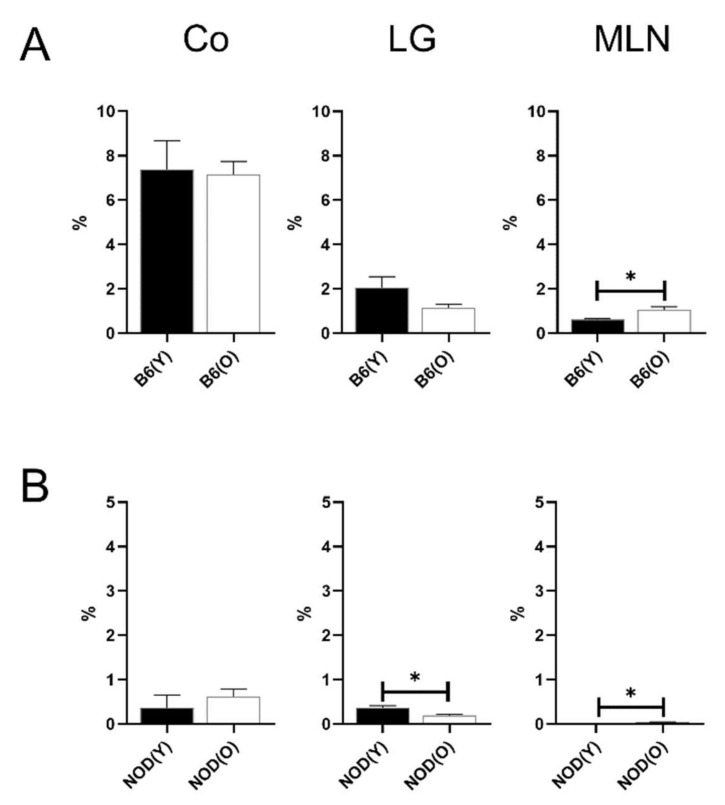
Distribution of neutrophil/eosinophils in B6 (**A**) (*n* = 4) and NOD.B10.H2^b^ (**B**) mice (*n* = 4). The total number of CD45^+^ cells was used as the denominator to compensate for the actual fraction of each subset in corneolimbus and LG. In MLN, the percentage of subsets was calculated as the percentage within total cells. The percentage of neutrophils/eosinophils (CD45^+^CD11b^+^CD11c^−^CD24^+^MHCII^lo^) was significantly higher in aged B6 mice than young B6 mice in MLN (*p* = 0.032). The percentage of neutrophils/eosinophils (CD45^+^CD11b^+^CD11c^−^CD24^+^MHCII^lo^) of LG was also lower in aged NOD.B10.H2^b^ mice than in young mice (*p* = 0.022). However, MLN showed a considerably increased proportion of neutrophils/eosinophils in aged NOD.B10.H2b mice (*p* = 0.015). All experiment results were tested for statistical differences using unpaired, two-tailed independent t-tests. Data are expressed as the mean ± standard error of the mean. B6, C57BL/6; NOD, NOD.B10.H2^b^; DC, dendritic cell; CO, corneolimbus; LG, lacrimal gland; MLN, mesenteric lymph node; B6(Y), young B6 mice; B6(O), aged B6 mice; NOD(Y), young NOD.B10.H2^b^ mice; NOD(O), aged NOD.B10.H2^b^ mice. * *p* < 0.05.

## Data Availability

All data generated or analyzed during this study are included in this published article.

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
