# Peer review of "Age-Dependent Distinct Distributions of Dendritic Cells in Autoimmune Dry Eye Murine Model"

_cells, 2021, doi:10.3390/cells10081857_

Round 1
Reviewer 1 Report
Cells-1263859
In this paper, authors examine the changes in distribution of dendritic cells in young and aged C57BL/6 (B6), as well as in NOD.B10.H2b mice. The formers develop dry-eye disease along their aging; the latter develop Sjögren’s syndrome-like characteristics. Authors show an analysis which shows that CD11b- and CD11b+ DCs, CD103+ DCs and MHC-IIhi B cells distribution in cornea, lacrimal gland and mesenterial lymph nodes is different for both C57BL/6 (B6) and NOD.B10.H2b mice. Interestingly, it is evident that young and aged NOD.B10.H2b mice undergo changes in dendritic cell distribution which are parallel to the aging process, as occurs in C57BL/6 (B6) mice. However there is a contrasting difference in NOD.B10.H2b compared to C57BL/6 (B6). Specifically, there is a predominance of CD11b + dendritic cells in the cornea, from aged individuals; a situation that is shown inverted in lacrimal gland and mesenterial lymph node. Also, a predominance of MHC-IIhi B cells was observed for aged NOD.B10.H2b aged mice in cornea, lacrimal gland and mesenterial lymph node in comparison with young mice; while in contrast for these B cells and for Neutrophils/eosinophils, differences between young and aged mice have no significance or were not so big. The manuscript is interesting and addresses an issue that is not so well studied.
MAJOR ISSUES:
- This study is merely descriptive. Interestingly this study shows that young and aged NOD.B10.H2b mice undergo changes in dendritic cell distribution which are parallel to the aging process, as seems to occur in C57BL/6 (B6) mice. Therefore, it would be necessary that authors carry out a much more careful discussion of the differences between both experimental models in order to have a clearer picture of the possible origins of such differences. The inflammatory process in DED in aged individuals is different from this Sjögren’s syndrome mimic? Even it would be preferable that authors make a direct comparison between both models in each figure of the manuscript.
- Authors should explain the reason to use a different staining procedure for C57BL/6 (B6) mice (fluorescein) vs NOD.B10.H2b mice (Lissamine Green B). While fluorescein staining dyes corneal damages, lissamine green B stains conjunctival tissue.
- Authors should indicate the antibody concentration and catalog number in order to enhance reproducibility of their results by others.
Author Response
I thank the editors and reviewers of “Cells”, for taking their time to review our manuscript titled as "Age-dependent distinct distributions of dendritic cells in auto-immune dry eye murine model". Having gone over the comments, the authors have made appropriate corrections and clarifications. The changes in responsive to reviewers’ comments have been described by red texts. Each of the co-authors has agreed with each of the changes made to this manuscript in the revision.
We hope that you will find our paper suitable for publication in your journal, and we look forward to hearing from you. I appreciate for your consideration of this revision.
----------------------------------------------------------------------------------------
Please check the pdf file attached (point-to-point responses to your comments).

Reviewer 2 Report
In this study, the profile of immune cells in the ocular surface, lacrimal gland and mesenteric lymph nodes of young and old B6 and NOD mice. The ocular surface integrity was measured clinically, and the cells quantified by flow cytometry. Few changes were observed in the corneal-limbal tissues in relation to immune cell profiles- but larger differences were observed in the number of CD11b+/- DCs in between young and old NOD mice. These findings may relate to higher severity of DED-like signs in old NOD mice.
- Overall the sample size is very low (n=4/group). Please justify this low sample size, and also include a rationale and justification for pooling both eyes in the statistical analyses. Regarding the presentation and statistical analysis of the data in Fig 2, it is unclear what each dot point on the graphs represents? If so, this needs to be reported, justified and appropriate statistical tests used. Number of animals should be reported in each figure legend.
- Please justify the inclusion of the mesenteric lymph node in the introduction or the methods. It is only on reading the discussion that it becomes (almost) clear why this LN was included (as a way of linking to gut dysbiosis?)
- The cornea and limbus contain very diverse populations of immune cells, with several immune cell subsets including T lymphocytes, macrophages, dendritic cells and NK cells residing in the limbus and conjunctiva. By pooling these tissue compartments with the cornea, the contribution of the corneal immune cells is effectively washed out. I suggest the term “cornea” is de-emphasised throughout the manuscript, and instead corneo-limbal or ocular surface be used instead.
- Figure 1 legend is highly repetitious to the description in the methods section.
- What is the rational for adjusting tear volume to body weight? Is there evidence that the Lacrimal Gland increases with body weight? Would this have been a better parameter to adjust tear secretion against?
- In Figure 5; the percentages of MHCII “MQ” and Neutrophil/eosinophil cells in the MLN are so low, it is unlikely to be biologically relevant. If the scale was adjusted to be consistent with the other panels, this would appear as almost zero. Also, the abbreviation of MQ for macrophages is unconventional, consider spelling out in full.
Author Response
I thank the editors and reviewers of “Cells”, for taking their time to review our manuscript titled as "Age-dependent distinct distributions of dendritic cells in auto-immune dry eye murine model". Having gone over the comments, the authors have made appropriate corrections and clarifications. The changes in responsive to reviewers’ comments have been described by red texts. Each of the co-authors has agreed with each of the changes made to this manuscript in the revision.
We hope that you will find our paper suitable for publication in your journal, and we look forward to hearing from you. I appreciate for your consideration of this revision.
Sincerely
----------------------------------------------------------------------------------------
Please check the pdf file attached (point-to-point responses to your comments).

Round 2
Reviewer 1 Report
Please check typing errors. Authors have fulfilled the concerns raised by reviewer. It could be convenient that authors carry out more extensive commentaries about the physiological implications or significance of their results, since this is an issue that can be improved.
Reviewer 2 Report
The authors have adequately responded to the Reviewers' concerns, and have appropriately included some of the study limitation in the discussion. The clarification of the revised manuscript has improved since the original submission.